# BUILDING THE DUAL GRAPH OF THE ACTIVATION REGIONS IN A DEEP NEURAL NETWORK: WHAT IT MEANS FOR GENERALIZATION

## ABSTRACT

Understanding the geometric representations of deep neural networks (DNNs) which employ a piecewise linear activation function has become a popular research direction for model explainability. A complete geometric picture of the representations of a DNN would include both the polytope regions formed by the network partitions and the set of neighboring regions, i.e., a dual graph. Prior work has resulted in algorithms which enumerate all of the activation regions formed by a network, but no algorithms have been proposed for constructing the dual graph in its entirety. This gap may stem from the naive assumption that because identifying neighboring regions is trivial in shallow networks, it is also trivial in deep networks. In this work, we demonstrate that this assumption is false; finding neighboring regions in a deep network is in fact a difficult problem due to the conditional nature of the partitions in the deep layers. We introduce a method to solve the difficult problem of neighbor finding in DNNs. We implement this algorithm along with region enumeration, which together constructs the dual graph. Further, we demonstrate the usefulness of the graph in the context of generalization. We show that test data that are near training data, as measured by path length along the graph, tend to yield the best generalization results.

## 1 INTRODUCTION

One promising approach to understanding why deep neural networks work so well, at least for networks with piecewise linear activation functions, is to look at the geometry of how the input space is partitioned into linear activation regions. Prior work has resulted in algorithms which enumerate the activation regions formed by a network (Chmielewski-Anders, 2020; Balestriero & LeCun, 2024; Robinson et al., 2020; Serra et al., 2018b). This approach has provided compelling evidence for relating the number and density of regions to important concepts, such as network expressivity (Hanin & Rolnick, 2019b;a; Montufar et al., 2014; Bianchini & Scarselli, 2014; Pascanu et al., 2013) and generalization (Novak et al., 2018; Humayun et al., 2024; Trimmel et al., 2021). However, are we exploiting the geometric structure of these networks to the greatest extent possible to answer questions about network performance? Here we argue no: a complete description of a network's geometric structure must go beyond enumeration of a network's activation regions; it must also reveal the underlying graph structure, i.e., construct the dual graph.

Although well-known mathematical constructs do not exist to characterize all neural networks, many common neural networks utilize continuous piecewise linear functions that can be written as spline operators (Balestriero et al., 2025; Huchette et al., 2023). Any such network behaves as follows: (1) the input space is partitioned into a set of convex polytope regions (called activation regions); (2) all the points within a polytope share a single affine transformation which generates the output. Therefore, if we want to understand the representations learned by an artificial neural network (ANN), we need to understand the underlying polytope structure instantiated by the network parameters. Specifically, we would like to know all the polytopes formed by the network, their properties, and how they tile the space.

While computing the dual graph for a network with a single hidden layer is trivial (since it amounts to finding the dual graph of a hyperplane arrangement), it is surprisingly difficult to compute the

dual graph of a deep neural network due to the conditional nature of the partitions in the higher layers. In this work, we demonstrate why finding graph edges is difficult in deep neural networks and provide the first exact algorithm (to the best of our knowledge) that constructs the dual graph of a DNN as well as approximate methods. We argue that there are three important benefits in constructing the dual graph: (1) the dual graph formalism provides a complete description of the geometric representation of such neural networks; (2) path length along the graph provides a distance metric more informative than Euclidean distance for assessing proximity of input vectors in a high dimensional space; and (3) test data that are near training data as measured by this distance metric tend to yield the best generalization results, a claim which we support with experiments.

## 2 RELATED WORK

Our algorithm for finding the dual graph of a neural network's activation regions builds on work in computational geometry for cell enumeration and finding tight hyperplanes in a hyperplane arrangement. For activation region enumeration in ANNs, there is an existing body of literature of which our implementation has similarities.

**Cell enumeration in hyperplane arrangements.** One of the most well known algorithms for cell enumeration is Avis and Fukuda's reverse search algorithm (Avis & Fukuda, 1996). We use a more recent algorithm (Rada & Černý, 2018), the incremental enumeration algorithm (IncEnum), which has been shown to outperform reverse search in practice.

**Finding the bounding hyperplanes of cells in hyperplane arrangements.** We apply known approaches for finding all tight hyperplanes for a cell in a hyperplane arrangement (Sleumer, 2000) with modifications for implementation convenience.

**Activation region enumeration in artificial neural networks.** Both approximate (Poole et al., 2016; Raghu et al., 2017; Novak et al., 2018; Hanin & Rolnick, 2019a; Serra & Ramalingam, 2020; Chmielewski-Anders, 2020) and exact (Chmielewski-Anders, 2020; Balestriero & LeCun, 2024; Robinson et al., 2020; Serra et al., 2018a) methods have been proposed for enumerating the activation regions in an ANN. A common exact enumeration approach is to identify regions by iterating through the network's layers from most shallow to most deep and partitioning the input space along the way. We leverage the IncEnum algorithm (Rada & Černý, 2018) to implement recursive partitioning, similarly to Balestriero & LeCun (2024).

## 3 BACKGROUND

In this section we provide brief summary of the background necessary for a geometric perspective of neural networks. For further reading, we refer the reader to Sleumer (2000) (Chapter 2), Balestriero et al. (2025), and Chmielewski-Anders (2020) (Chapter 2).

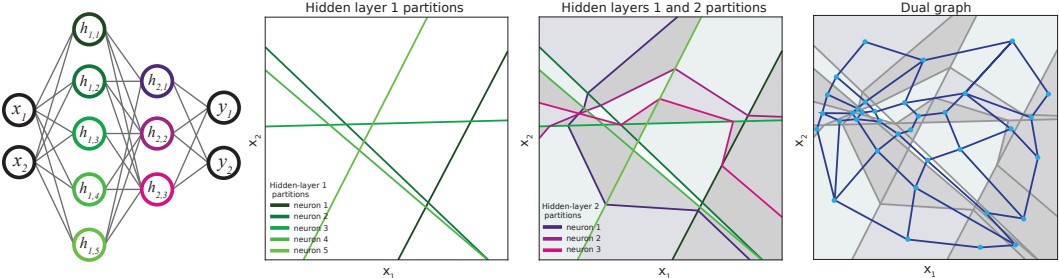

Figure 1: An example of how a neural network partitions its input space into activation regions, and the corresponding dual graph. Left: Example network. Middle-left: The hyperplanes formed by the neurons in the first hidden layer of the network. Middle-right: The partitions formed by the neurons in both the first (green) and second (pink/purple) hidden layers of the network. Right: Network activation regions, and the corresponding dual graph.

## 3.1 DEEP NEURAL NETWORKS

For simplicity, we consider multilayer perceptron networks with $L \geq 1$ hidden layers that employ the rectified linear unit (ReLU) activation function, $[x]^+ = \max\{0, x\}$. The network takes inputs $\mathbf{x} \in \mathbb{R}^d$ and produces outputs $\mathbf{y} \in \mathbb{R}^m$. Each hidden layer $l$ has $n_l$ neurons, and the total number of hidden layer neurons is $n = \sum_l n_l$. Each neuron computes a linear transformation followed by ReLU to produce the layer activations $\mathbf{h}_l = [h_{l,1}, h_{l,2}, ..., h_{l,n_l}]^T$. The transformation, $f_l : \mathbb{R}^{n_{l-1}} \to \mathbb{R}^{n_l}$, is parameterized by a weight matrix, $\mathbf{W}_l \in \mathbb{R}^{n_l \times n_{l-1}}$, and a bias vector, $\mathbf{b}_l \in \mathbb{R}^{n_l}$. The network's output layer is a linear transformation: $\mathbf{y} = \mathbf{W}_{L+1}\mathbf{h}_L + \mathbf{b}_{L+1}$. Together these transforms construct a piecewise affine mapping, $F : \mathbb{R}^d \to \mathbb{R}^m$, by partitioning the input space of the network into a set of convex regions where points within a region share a single affine transformation. Region boundaries result from neurons in a hidden layer of the network switching from positive (zero) to zero (positive). We refer to these convex regions as activation regions (Figure 1).

**Definition 3.1** (Activation pattern). For a given ANN, the corresponding *activation pattern* of input $\mathbf{p} \in \mathbb{R}^d$ is a indicator vector $\mathbf{z} = \langle \mathbf{z}_1, ..., \mathbf{z}_L \rangle$ where $\mathbf{z}_l$ is itself an indicator vector with values:

$$z_{l,i} = \begin{cases} 1 & \text{if } h_{l,i} > 0 \\ 0 & \text{if } h_{l,i} \leq 0 \, . \end{cases}$$

**Definition 3.2** (Activation region). For a given ANN, an *activation region* is the set of points $\mathbf{p} \in \mathbb{R}^d$ with the same *activation pattern* $\mathbf{z}$.

## 3.2 HYPERPLANE ARRANGEMENTS

ANNs form activation regions by partitioning the input ambient space, $\mathbb{R}^d$, via hyperplanes into cells.

**Definition 3.3** (Hyperplane arrangement). A *hyperplane arrangement*, $\mathcal{A}$, is a finite set of distinct hyperplanes, where a hyperplane is the set $\{\mathbf{x} \in \mathbb{R}^d : \mathbf{a}^T\mathbf{x} = b\}$, where $\mathbf{a} \in \mathbb{R}^d \setminus \{0\}$ and $b \in \mathbb{R}$. A hyperplane arrangement partitions $\mathbb{R}^d$ into convex polyhedral regions called faces.

**Definition 3.4** (Cell). A *cell*, $C$, is a $d$-dimensional face formed by a hyperplane arrangement.

**Definition 3.5** (Sign vector). Any point $\mathbf{p} \in \mathbb{R}^d$ can be described by its relationship to each hyperplane in $\mathcal{A}$. Let $H_i = \{\mathbf{x} \in \mathbb{R}^d : \mathbf{a}^T\mathbf{x} = b\}$. If $\mathbf{a}^T\mathbf{p} > b$, then $\mathbf{p}$ is on the positive side of hyperplane $H_i$, denoted as $H_i^+$. Similarly, if $\mathbf{a}^T\mathbf{p} < b$, then $\mathbf{p}$ is on the negative side of the hyperplane, denoted as $H_i^-$. If $\mathbf{a}^T\mathbf{p} = b$, the point $\mathbf{p}$ falls on the hyperplane $H_i$. We describe the location of $\mathbf{p}$ by a *sign vector*, $\mathbf{s}$, consisting of $+$, $0$, and $-$ as follows:

$$s_i = \begin{cases} + & \text{if } \mathbf{p} \in H_i^+ \, , \\ 0 & \text{if } \mathbf{p} \text{ is on } H_i \, , \\ - & \text{if } \mathbf{p} \in H_i^- \, . \end{cases}$$

Points that have the same sign vector can be grouped into *faces*.

Notice the sign vector of a cell consists solely of $+$'s and $-$'s. All points $\mathbf{p} \in C$ have the same vector. Let this sign vector be $\mathbf{s}$. We say that cell $C(\mathbf{s})$ is described by $\mathbf{s}$.

## 3.3 REGION PARTITIONING IN NEURAL NETWORKS

The first layer of a neural network has activations $\mathbf{h}_1 = \mathbf{W}_1\mathbf{x} + \mathbf{b}_1$ for input $\mathbf{x}$. Each neuron forms a hyperplane $H_{1,i} = \{\mathbf{x} \in \mathbb{R}^d : \mathbf{w}_{1,i}\mathbf{x} = -b_{1,i}\}$ in the input space of the network at the non-differentiable boundary of the ReLU activation function. Together these hyperplanes form an arrangement and each cell corresponds to a different affine transformation; the affine transformation is determined by the neurons that are active within the cell (Figure 1 middle-left).

The activation of neurons in deep layers of the network are conditioned on the location of $\mathbf{x}$ with respect to the affine transformations formed by the previous layers. We can write the activity of hidden layers $l \geq 2$ as follows[1]:

---

[1]The ReLU is applied element-wise.

$$\mathbf{h}_2 = \mathbf{W}_2[\mathbf{W}_1\mathbf{x} + \mathbf{b}]^+ + \mathbf{b}_2$$
$$= \mathbf{W}_2\mathbf{D}_1^{(\mathbf{x})}\mathbf{W}_1\mathbf{x} + \mathbf{D}_1^{(\mathbf{x})}\mathbf{b}_1 + \mathbf{b}_2$$
$$= \mathbf{W}_2^*\mathbf{x} + \mathbf{b}_2^*$$
$$\mathbf{h}_l = \mathbf{W}_l\mathbf{D}_{l-1}^{(\mathbf{x})}\mathbf{W}_{l-1}^*\mathbf{x} + \mathbf{W}_l\mathbf{D}_{l-1}^{(\mathbf{x})}\mathbf{b}_{l-1}^* + \mathbf{b}_l$$
$$= \mathbf{W}_l^*\mathbf{x} + \mathbf{b}_l^*,$$

where $\mathbf{D}_l^{(\mathbf{x})}$ is a diagonal matrix with the $l$-th layer indicator vector $\mathbf{z}_l^{(\mathbf{x})}$ along the diagonal for input $\mathbf{x}$. $\mathbf{W}_l^*$ and $\mathbf{b}_l^*$ are the effective weights and biases, respectively, for input $\mathbf{x}$. For all inputs that fall within the same cell formed by layers 1 to $l-1$, the effective weights and biases will be the same. All transformations within each cell will have contributions from a differing sets of neurons, and thus the effective weights matrices and bias vectors will differ.

Given a cell formed by layers 1 to $l-1$, we can form a hyperplane arrangement from the neurons in layer $l$ using $\mathbf{W}_{l-1}^*$ and $\mathbf{b}_{l-1}^*$. Some of these hyperplanes will intersect the cell, further partitioning the space and introducing new nonlinearities. Other hyperplanes will lie completely outside of the cell, contributing a linear term to the transformation function within the cell. Because the ReLU activation function is continuous, a partitioning hyperplane must form a continuous partition across all cells. The changing effective weights and bias parameters across cells allows hyperplanes from neurons in deep layers to bend at cell boundaries (Figure 1middle right).

At the final layer, all partitions have been applied. Each cell corresponds to an activation region of the network, and all points in the input space can be describe in relation to the orientation of all hyperplanes via a *network sign vector*.

**Definition 3.6** (Network sign vector). For a given ANN, the corresponding *network sign vector* (NSV) of input $\mathbf{p} \in \mathbb{R}^d$ is the vector $\mathbf{v} = \langle \mathbf{s}_1, ..., \mathbf{s}_L \rangle$, where $\mathbf{s}_l$ is the sign vector of the hyperplane arrangement formed by the $l$-th layer of the network, with respect to $\mathbf{p}$.

If $\mathbf{p}$ falls within a cell, then from Definition 3.1, $\mathbf{p}$ can be described as a NSV with only '$+$' and '$-$' values. We say an input $\mathbf{p}$ corresponds to NSV $\mathbf{v}$ if $\mathbf{p}$ is within the face defined by $\mathbf{v}$. All points within the cell $C(\mathbf{v})$ have the same NSV $\mathbf{v}$. Thus, a deep neural network forms a partition of cells, known as activation regions, each with a unique NSV.

## 4 PROBLEM

### 4.1 THE NEURAL NETWORK DUAL GRAPH PROBLEM

We are interested in representing the partition structure of a given neural network as a simple graph where vertices are activation regions and edges connect neighboring regions (Figure 1 right); we call this the ANN dual graph problem. The problem can be split into two distinct subproblems: 1) enumerating the activation regions, i.e., finding the graph's vertices, and 2) finding the regions that neighbor each other, i.e., finding the edges of the graph.

**The ANN dual graph problem.** The problem takes as input the parameters of a neural network $\theta = \cup_{l=1}^{L+1}\{\mathbf{W}_l, \mathbf{b}_l\}$, and outputs the dual graph $G = \{V, E\}$ where $V$ is the set of all activation regions, and $E$ is the set of all neighbor pairs. There are many ways to refer to an activation region. In this paper, we will use network sign vectors as reference ids.

**The activation region enumeration subproblem.** The activation region enumeration problem takes as input $\theta$ and outputs $V$.

**The neighbor finding subproblem.** The neighbor finding problem takes as input $\theta$ and $V$. The output is a set containing all region pairs:

$$E = \big\{\{\mathbf{v}, \mathbf{u}\} \mid \mathbf{v}, \mathbf{u} \in V, \ \mathbf{v} \text{ and } \mathbf{u} \text{ are neighbors}\big\}.$$

By solving both subproblems, one can construct the dual graph.

For simplicity, we have presented the problem statement here for an unbounded input domain. For a bounded domain, our algorithms presented in Section 5 can be adjusted as follows. Let the convex bounded input domain be $\mathbb{X}$. We can construct a set of hyperplanes, $X$, where the union of the half spaces of the hyperplanes in $X$ is $\mathbb{X}$. Now, we include the hyperplanes of $X$ as additional constraints in the linear program calls of our algorithms. In other words, if an activation region lies outside of the bounded domain, it will not be found because the constraints of $X$ will not be satisfied.

### 4.2 The challenge of neighbor finding

Given $V$, finding neighbors in a neural network with one hidden layer is trivial, as this amounts to finding neighboring cells in a hyperplane arrangement. In a hyperplane arrangement, cells are neighbors if and only if their sign vectors have a Hamming distance of 1. Thus, finding the edges of the dual graph of a shallow network is as simple as identifying all pairs of NSVs with a Hamming distance of 1.

However, in DNNs there is *one-sided correctness*: all neighbors have NSVs of Hamming distance 1, but not all regions with NSVs of Hamming distance 1 are neighbors. Thus, we explicitly identify tight hyperplanes to correctly determine neighboring regions.

The difficulty arises from the conditional nature of deep layer partitions. Neurons in deep layers create partitions that can bend throughout the input space, making it possible for two regions to have NSVs with a Hamming distance of 1, but not be remotely close to neighboring one another. For example, this could happen if a partition bends back around on itself as demonstrated in Figure 2.

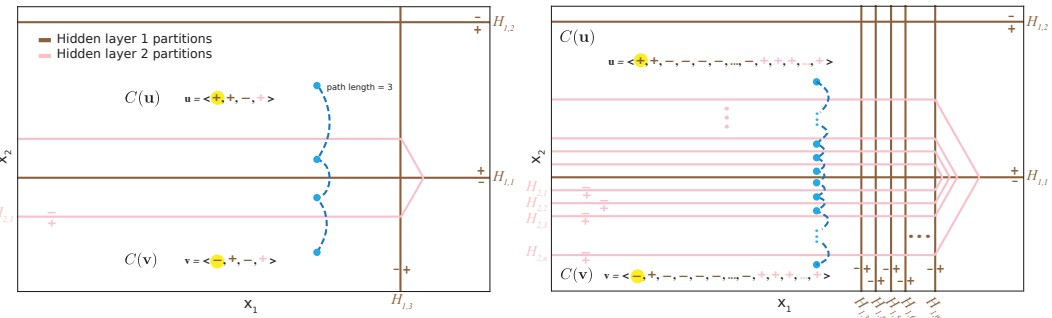

Figure 2: Example DNNs where regions $C(\mathbf{u})$ and $C(\mathbf{v})$ have sign vectors with Hamming distance 1 but are not neighbors. Left: A network with $n_1 = 3$ (brown hyperplanes) and $n_2 = 1$ (pink hyperplane). Network sign vectors $\mathbf{v}$ and $\mathbf{u}$ differ in only the sign of hyperplane $H_{1,1}$ (yellow circle), but the shortest path in the dual graph is of length 3 (blue path). Right: A network can be constructed such that the shortest path (blue path) between regions $C(\mathbf{u})$ and $C(\mathbf{v})$ is arbitrarily large despite $\mathbf{u}$ and $\mathbf{v}$ having a Hamming distance of 1 (see Appendix A for the network parameters).

In the left panel of Figure 2 there are three neurons in the first hidden layer (brown hyperplanes) and one neuron in the second hidden layer (pink hyperplane). Because of the bend formed by the second hidden layer neuron, the two regions $C(\mathbf{u})$ and $C(\mathbf{v})$ have a shortest path length of three in the dual graph despite having a Hamming distance of 1. In fact, one can construct a network with arbitrarily large error between path distance and Hamming distance (Figure 2 right).

## 5 Algorithm

We present an algorithm that constructs the dual graph of a DNN by sequentially solving the activation region enumeration and neighbor finding subproblems. To the best of our knowledge, this is the first algorithm for finding neighbors.

The two steps of the algorithm are as follows. First, the set of activation regions, $V$, is found by an iterative partitioning process of the input space from shallow to deep layers. Second, the neighbors of each region $\mathbf{v} \in V$ is found by identifying the neurons whose corresponding partitions bound $C(\mathbf{v})$. The outputs from both steps together form the dual graph, $G = \{V, E\}$.

Our algorithm is output-polynomial (i.e., the running time is polynomial in the size of the output; Appendix D) – both region enumeration and neighbor finding are polynomial in $V$. Note, the size of $V$ can vary widely depending on the values of $\theta$.

## 5.1 Activation region enumeration

Our approach for region enumeration is similar to existing works in the literature (Robinson et al., 2020; Balestriero & LeCun, 2024). Balestriero & LeCun (2024) presented an algorithm similar to IncEnum for finding cells in network hyperplane arrangements; they implemented and tested their algorithm on shallow networks leveraging parallel computations. Here we extend their work, explicitly implementing the IncEnum algorithm and applying it to deep networks. Our algorithm takes as inputs the parameters, $\theta$, of a given neural network and returns the set, $V$, of all NSVs.

For a comprehensive understanding of existing iterative region enumeration approaches, we point the reader to the works mentioned above. For a detailed description of our region enumeration approach and the computational geometry methods it builds upon see Appendix B.

## 5.2 Neighbor finding

Here we present our algorithm for identifying neighboring activation regions in the input space tessellation of a DNN. Given the set of NSVs, $V$, and network parameters, $\theta$, our algorithm (`NN_Neighbors`; Algorithm 5 in Appendix C) identifies, for each region $\mathbf{v} \in V$, its neighbors via a direct application of methods from computational geometry for finding tight hyperplanes of a cell in a hyperplane arrangement (Sleumer, 2000).

There are four main steps to our approach. First, for each $\mathbf{v}$, form a hyperplane arrangement, $\mathcal{A}_{\mathbf{v}}$, with a hyperplane for each neuron in the network parameterized by its effective weights and bias with respect to the points in region $C(\mathbf{v})$ (we call this a *network hyperplane arrangement*; Definition 5.1). Next, find an interior point $\mathbf{p}$ of cell $C(\mathbf{v})$ in $\mathcal{A}_{\mathbf{v}}$. Third, find an initial set of tight hyperplanes that define a vertex of $C(\mathbf{v})$. Finally, find all remaining tight hyperplanes.

After all tight hyperplanes have been found, we can identify the neighboring regions. For each tight hyperplane, flip the sign of the corresponding element in the NSV $\mathbf{v}$ to get $\mathbf{v}'$. The pair $\{\mathbf{v}, \mathbf{v}'\}$ is an edge in $G$.

Below we explain the steps of our algorithm in detail. For simplicity, we will assume that all hyperplane arrangements are in general position. We have not found this assumption to be a problem in our implementation and experiments.[2]

### 5.2.1 Constructing the network hyperplane arrangement

For any point in the input space of a network, $\theta$, we can form a *network hyperplane arrangement*.

**Definition 5.1** (Network hyperplane arrangement). For a point, $\mathbf{x} \in \mathbb{R}^d$, there is a hyperplane arrangement

$$\mathcal{A}_{\mathbf{v}} = \left\{ \mathbf{x} \in \mathbb{R}^d : \mathbf{w}_{l,i}^* \mathbf{x} + b_{l,i}^* = 0 \mid 1 \leq l \leq L, 1 \leq i \leq n_l \right\},$$

where $\mathbf{w}_{l,i}^*$ and $b_{l,i}^*$ are the effective weights and bias for the $i$th neuron in the $l$th layer, respectively.

All points in the input space within an activation region have the same effective weights and bias parameters, and thus will form the same network hyperplane arrangement. The tight hyperplanes of the cell in the arrangement are the same neurons that bound the activation region in the network's tessellation of the input space. In other words, if we can find the tight hyperplanes of $C(\mathbf{v})$ in $\mathcal{A}_{\mathbf{v}}$, we have found the neurons that differentiate $C(\mathbf{v})$ from its neighboring activation regions. So, the first step, `NetHypArr`, of our algorithm constructs $\mathcal{A}_{\mathbf{v}}$ from $\theta$ and $\mathbf{v}$ (Figure 3 top-left and top-middle) via the effective weights and biases.

### 5.2.2 Finding an interior point

The second step is to find an interior point, $\mathbf{p}$, of $C(\mathbf{v})$, which is needed for finding the tight hyperplanes. There are a variety of methods that work. Our procedure, `IntPt`, finds point $\mathbf{p}$ by

---

[2]If it becomes a problem, see Sleumer (2000) for solutions.

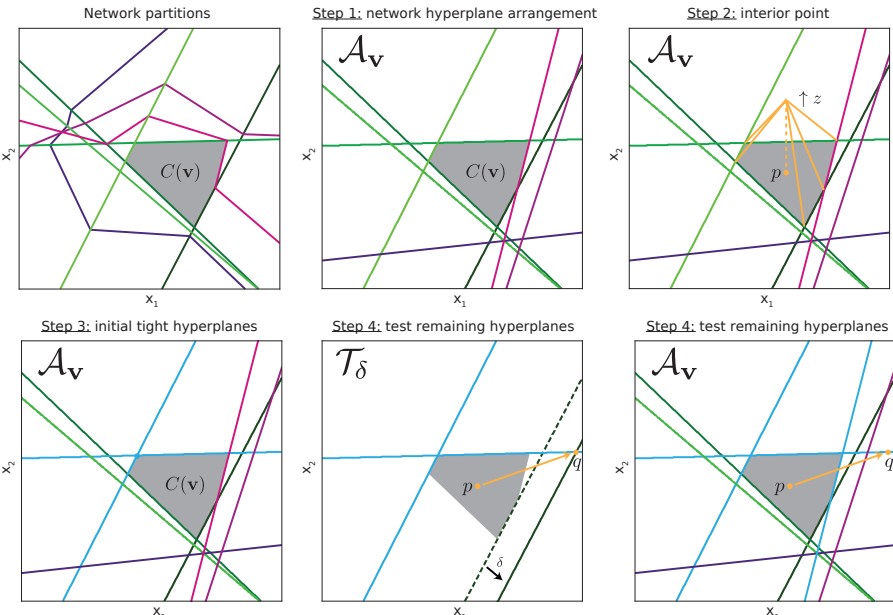

Figure 3: An illustration of the neighbor finding algorithm. Network and neuron partition colors are the same as in Figure 1. Top-left: Activation region $C(\mathbf{v})$ in the network's tessellation of the input space. Top-middle: Network hyperplane arrangement, $\mathcal{A}_{\mathbf{v}}$, for points in $C(\mathbf{v})$. Top-right: Interior point finding for cell $C(\mathbf{v})$ in $\mathcal{A}_{\mathbf{v}}$. Bottom-left: An initial set of tight hyperplanes (cyan) from a defining vertex of $C(\mathbf{v})$ in $\mathcal{A}_{\mathbf{v}}$. Bottom-middle: Perturbation of selected hyperplane to test for inclusion ($H_{1,1}$), arrangement $\mathcal{T}_{\delta}$ to find $\mathbf{q}$, and ray $\overline{\mathbf{pq}}$ (orange). Bottom-right: The first hyperplane not in $\mathcal{T}$ intersected by $\overline{\mathbf{pq}}$ is $H_{2,3}$, not $H_{1,1}$. Include $H_{2,3}$ in the set of tight hyperplanes (cyan).

maximizing $z$ subject to the constraints of the following linear program [3]:

$$
\begin{aligned}
(\mathbf{w}_{l,i}^{*}\mathbf{x} + b_{l,i}^{*}) - z \geq 0 \quad & \forall H_{l,i} \in \mathcal{A}_{\mathbf{v}}^{+} \\
-1(\mathbf{w}_{l,i}^{*}\mathbf{x} + b_{l,i}^{*}) - z \geq 0 \quad & \forall H_{l,i} \in \mathcal{A}_{\mathbf{v}}^{-} \\
z \leq 1 \quad & \text{(or any other positive constant),}
\end{aligned}
$$

where $H_{l,i} \in \mathcal{A}_{\mathbf{v}}^{+}$ if $v_{l,i}$ is '+', otherwise $H_{l,i} \in \mathcal{A}_{\mathbf{v}}^{-}$ and $v_{l,i}$ is '-'. All hyperplanes are oriented to face towards $C(\mathbf{v})$, and we maximize over a newly introduced dimension, $z$.

Intuitively, one can think of this as optimizing towards the apex of a "pyramid" on top of $C(\mathbf{v})$. The projection of apex back onto $C(\mathbf{v})$ is the returned interior point, $\mathbf{p}$ (Figure 3 top-right). The interior point does not need to be near the centroid, and thus $z$ can be upper-bounded by any positive constant.

### 5.2.3 FINDING AN INITIAL SET OF TIGHT HYPERPLANES

Before we can find the set of all tight hyperplanes, we first construct an initial set of tight hyperplanes, $\mathcal{T}_{0}$, by calling our procedure `InitTightHyp`. The hyperplanes in $\mathcal{T}_{0}$ must be vertex defining, meaning that their intersection is a vertex in $C(\mathbf{v})$. To identify them, we first select any vertex of $C(\mathbf{v})$ by finding a basic feasible solution to the linear constraints:

$$
\begin{aligned}
\mathbf{w}_{l,i}^{*}\mathbf{x} + b_{l,i}^{*} \geq 0 \quad & \forall H_{l,i} \in \mathcal{A}_{\mathbf{v}}^{+} \\
-1(\mathbf{w}_{l,i}^{*}\mathbf{x} + b_{l,i}^{*}) \geq 0 \quad & \forall H_{l,i} \in \mathcal{A}_{\mathbf{v}}^{-}.
\end{aligned}
$$

Next we check which hyperplanes in $\mathcal{A}_{\mathbf{v}}$ intersect with the found vertex.[4] These hyperplanes are elements of $\mathcal{T}_{0}$ (Figure 3 bottom-left, blue hyperplanes).

---

[3]This is the same method used in Sleumer (2000) Section 3.2.1.

[4]This works because we assume that all hyperplanes are in general position.

### 5.2.4 FINDING THE REMAINING TIGHT HYPERPLANES

Next we test if any of the remaining hyperplanes, $\mathcal{U} := \mathcal{A}_\mathbf{v} \setminus \mathcal{T}_0$, are tight to $C(\mathbf{v})$. The procedure, `TightHyp` (Algorithm 6 in Appendix C), for finding the remaining hyperplanes is as follows.

Let $\mathcal{T} = \mathcal{T}_0$. Pick a hyperplane $H_{l,i} \in \mathcal{U}$. First, we will test if $H_{l,i}$ is tight to $\mathcal{T}$. Orient all hyperplanes in $\mathcal{T} \cup H_{l,i}$ to face the interior point $\mathbf{p}$ of $C(\mathbf{v})$. Perturb $H_{l,i}$ a distance $\delta$ in the perpendicular direction away from $\mathbf{p}$. Call this new arrangement $\mathcal{T}_\delta$ (Figure 3 bottom-middle). Next, find a witness point, $\mathbf{q}$ that is both on $H_{l,i}$ and a vertex in $\mathcal{T}_\delta$; do this by optimizing in the direction of $H_{l,i}$ with a linear program. If no such witness exists, then $H_{l,i}$ cannot possibly be tight to $C(\mathbf{v})$.

Let the predicate $W_\mathbf{q}(H_{l,i})$ be true if the witness $\mathbf{q}$ exists, and false otherwise. If $W_\mathbf{q}(H_{l,i})$ is true, shoot a ray, $\overline{\mathbf{pq}}$, from $\mathbf{p}$ to $\mathbf{q}$ (Figure 3 bottom-middle), and find the first hyperplane $H_{k,j} \in \mathcal{U}$ that intersects it. $H_{k,j}$ is a tight hyperplane of $C(\mathbf{v})$ (Figure 3 bottom-right). Add $H_{k,j}$ to $\mathcal{T}$ and remove $H_{k,j}$ from $\mathcal{U}$. If $W_\mathbf{q}(H_{l,i})$ is false, remove $H_{l,i}$ from $\mathcal{U}$.

Continue this process, selecting new hyperplanes from $\mathcal{U}$ until the set is empty. Note, the hyperplane being tested, $H_{l,i}$, may not be the hyperplane, $H_{k,j}$, that is found to be tight. If this is the case we cannot remove $H_{l,i}$ from $\mathcal{U}$, as it could still be a tight hyperplane. When $\mathcal{U}$ is empty, all tight hyperplanes have been found.

### 5.2.5 COMPLETING THE DUAL GRAPH

The tight hyperplanes of $C(\mathbf{v})$ in $\mathcal{A}_\mathbf{v}$ separate the activation region from its neighboring regions in the input space tessellation formed by the DNN. By finding the tight hyperplanes of $C(\mathbf{v})$, we have found the sign vectors for all neighboring activation regions. For each tight hyperplane $T_{l,i} \in \mathcal{T}$, we can flip the sign of $i$-th neuron in the $l$-th layer to get the sign vector, $\mathbf{v}'$, of the neighboring activation region, i.e., $\mathbf{v}' := \mathbf{v}$ where $v'_{l,i} = -v_{l,i}$. The pair $\{\mathbf{v}, \mathbf{v}'\}$ is an edge in G.

For a correctness argument see Appendix E.

## 6 EXPERIMENTS

We analyzed the dual graph in the context of generalization as a first step in demonstrating its usefulness in model explainability. Specifically, for a region occupied by a test point, we computed the shortest path in the dual graph to regions occupied by training data; we call the length of the shortest path between two activation regions the *bin distance*. We also found the Hamming distance of activation region NSVs to be a good approximation of the bin distance, enabling us to extend our generalization analysis to a larger network. For implementation details and validation on a toy problem see Appendix F.

### 6.1 DUAL GRAPH PATH LENGTH AND GENERALIZATION

To analyze bin distance, we trained three networks to solve a 15-class subset of the Extended MNIST (EMNIST-15) image classification problem (Cohen et al., 2017) with 2,400 training and 400 testing samples per class. The networks were trained using Adam optimizer (Kingma & Ba, 2014) with learning rate $10^{-3}$ and batch size 128. Networks were initialized with i.i.d. normal weights and biases with variance 2/fan-in and $10^{-6}$, respectively (Hanin & Rolnick, 2019a). All networks had two hidden layers both of width 11, and the input images were scaled to 14x14. After training, we ran our algorithm to construct the dual graph for each network. The number of vertices (i.e., number of activation regions), number of edges, average vertex degree, and serial runtime of both steps of the algorithm are reported in Table 1.

We tested if the graph structure was predictive of network generalization ability to fit unseen test data. For each test point, we computed the bin distance from the vertex occupied by the point to the nearest vertex occupied by a correctly classified training point of the same class label. For each incorrectly classified test point, we further calculated the bin distance from its vertex to the nearest vertex occupied by a correctly classified training point of the *predicted* class label.

Table 1: Graph properties and serial runtimes for networks trained on the EMNIST-15 problem.

| Network | $|V|$ | $|E|$ | Avg. degree +/- std. | Region enum. | Neighbor finding |
|---|---|---|---|---|---|
| | | **Graph properties** | | **Empirical runtime (hrs)** | |
| Network 1 | 609,859 | 4,869,114 | 15.14 +/- 2.89 | 1.7228 | 131.2093 |
| Network 2 | 589,470 | 4,707,404 | 15.05 +/- 2.75 | 1.5240 | 226.9693 |
| Network 3 | 365,193 | 2,837,973 | 14.74 +/- 2.77 | 1.0757 | 221.2067 |

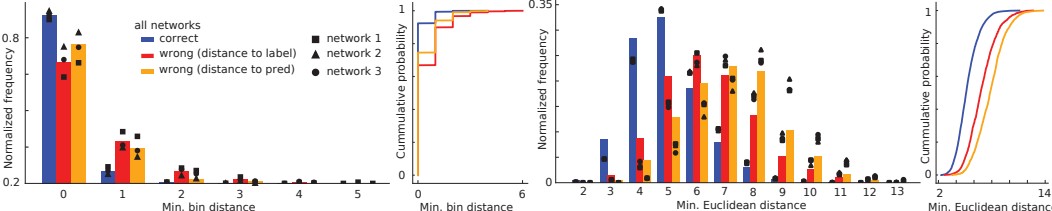

Figure 4: Generalization in terms of bin distance in the dual graph of the same three models from Table 1. Left and left-middle: Shortest path in the dual graph of correctly classified test points to correctly classified training points of the same class label (blue), incorrectly classified test points to correctly classified training points of the same class label (red), and incorrectly classified test points to correctly classified training points of the predicted class label (orange). Incorrectly classified test points are closer to predicted label than true class label. Right-middle and right: Same as left plots but for Euclidean distance in the input space. Incorrectly classified test points are closer to true class label than predicted label.

We found that path lengths of correctly classified test points were shorter than those of incorrectly classified points; this was true of Euclidean distance in the input space as well. Interestingly, wrongly classified test points were closer to their predicted class label in the graph but not in the input space (Figure 4).

## 6.2 APPROXIMATE NEIGHBOR FINDING AND BIN DISTANCE

To extend our analysis to larger networks, we applied approximate methods. We evaluated the accuracy of an approximate neighbor finding method and an approximate bin distance method in our networks trained on EMNIST-15. We then applied the approximate bin distance method to a convolutional neural network (CNN) trained on CIFAR-10.

To approximately identify neighbors, we use a naive approach that assumes all regions with NSVs of Hamming distance 1 are in fact neighbors (Section 4.2). Thus finding the neighbors of $C(\mathbf{v})$, is a simple iteration over the elements of $\mathbf{v}$. For each element in $\mathbf{v}$ flip its sign to get $\mathbf{u}$ and test if $\mathbf{u}$ is in $V$. In our networks trained on EMNIST-15, we found very few aberrant edges when using this method (Table 2 left).

We next analyzed if the Hamming distance between two NSVs was a good approximation of bin distance between the corresponding regions. For each network trained on EMNIST-15, we computed the difference between the Hamming distance and bin distance for both the pairs used in the bin distance generalization analysis and 100,000 random pairs. We found no differences (Table 2 right).

The high accuracy of Hamming distance as an approximate bin distance metric enabled us to apply our generalization analysis to a larger network, bypassing the construction of the dual graph. We trained a CNN with 5 convolutional layers, two fully connected layers, and a linear readout layer on the CIFAR-10 dataset (similar to Humayun et al. (2024)). Using Hamming distance instead of bin distance, we applied the same generalization analysis as previously done on the networks trained on EMNIST-15. We found the Hamming distance of correctly classified test points was smaller than those of incorrectly classified points; this was no longer true of Euclidean distance in the input space (Figure 5 left and middle). We no longer found wrongly classified test points to be closer to their predicted class label than true class label; both distances were similar. As expected in high

| | Approx. neighbors | | Hamming dist. (# errs. / # region pairs) | |
|---|---|---|---|---|
| **Network** | Num. errors | Error rate ($\times 10^{-4}$) | Random region pairs | Figure 4 pairs |
| Network 1 | 513 | 1.0535 | $0/10^5$ | 0/6878 |
| Network 2 | 311 | 0.6606 | $0/10^5$ | 0/6850 |
| Network 3 | 319 | 1.1239 | $0/10^5$ | 0/6837 |

Table 2: Empirical error of approximate methods of neighbor finding and bin distance.

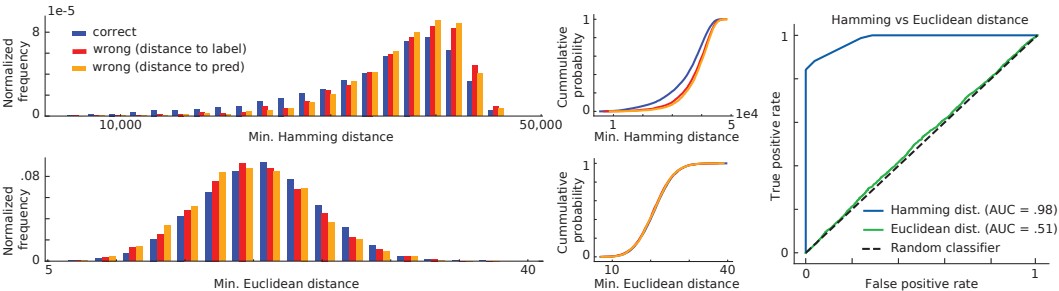

Figure 5: Generalization using Hamming distance between activation regions in a CNN model trained on CIFAR-10. Left-top: Same as Figure 4 for Hamming distance. Left-bottom: Euclidean distance. Right: ROC curves for Hamming and Euclidean distance.

dimensions, Euclidean distance was an uninformative metric. Interestingly, Hamming distance was a stronger predictor of generalization (Figure 5 right).

# 7 DISCUSSION

We introduced a method to find the dual graph of the input space partitions formed by a neural network. We show that while neighbor finding is trivial for a single layer network, it is not so for deep networks, where NSVs having Hamming distance of 1 between activation regions is necessary but not sufficient for adjacency. Thus, this work primarily focuses on the algorithm necessary to compute the exact dual graph. The dual graph completely describes the network's geometric representation. We demonstrated its utility for model explainability and generalization, showing that the shortest bin distances between test and training data successfully predicted the network's ability to generalize.

This dual graph structure provides a principled approach to measure distance without succumbing to the ill-defined problem of extrapolation or interpolation for data in high dimensions (Balestriero et al., 2021), which is impossible with region enumeration alone. In fact, Balestriero et al. (2018) analyzed feature similarity with respect to approximate bin distances in CNNs trained on CIFAR10 and SVHN. They found that for inputs closer in bin distance, both feature similarity and Euclidean distance grew as network depth increased.

The dual graph formalism presents several promising directions for future research. First, the observed correlation between bin distance and generalization should be explored in greater detail by linking it to the image features discussed in Balestriero et al. (2018). Second, the formalism enables a structural analysis where features are represented as cuts and class boundaries as walks in the graph, offering a unique way to understand the stability of features and the network's decision boundaries. Third, it is crucial to compare bin distance against established generalization metrics, such as the input-output Jacobian (Novak et al., 2018), to validate the approach and clarify its unique insights. Finally, the dual graph can be leveraged to understand generalization and adversarial examples by analyzing the piecewise linear structure of neighborhoods in the graph surrounding input points and by using bin distance to identify the minimal structural changes required to cross a decision boundary.

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

## A    NON-NEIGHBORS EXAMPLE

Below are the network parameters used to produce right plot in Figure 2. The ellipses represent that any number of new neurons could be introduced into the network to make the error between Hamming distance and shortest path in the dual graph between regions $C(\mathbf{v})$ and $C(\mathbf{u})$ arbitrarily large.

$$
\mathbf{W}_1 = \begin{bmatrix} 1 & 0 \\ 0 & 1 \\ 0 & -1 \\ 1 & 0 \\ 1 & 0 \\ 1 & 0 \\ & \vdots \\ 1 & 0 \end{bmatrix}, \mathbf{b}_1 = \begin{bmatrix} -3 \\ 0 \\ 6 \\ -.5 \\ -1 \\ -1.5 \\ \vdots \\ -2 \end{bmatrix}, \mathbf{W}_2 = \begin{bmatrix} 1 & 1 & .5 & 0 & 0 & 0 & 0 \\ 1 & 1 & .5 & 0 & 0 & 0 & 0 \\ 1 & 1 & .5 & 0 & 0 & 0 & 0 \\ & & & \vdots & & & \\ 1 & 1 & .5 & 0 & 0 & 0 & 0 \end{bmatrix}, \mathbf{b}_2 = \begin{bmatrix} -3 \\ -3.25 \\ -3.5 \\ -3.75 \\ \vdots \\ -4.5 \end{bmatrix}
$$

## B    CELL ENUMERATION ALGORITHM

In this section, we describe our algorithm to enumerate all of the activation regions of a neural network. The method is a layer-wise process. At a given layer $l$, for each cell formed by layers 1 to $l-1$, we find how the neurons in $l$ further partition the cell. We do this by constructing the hyperplane arrangement of layer $l$ with the effective weights and biases for the given cell (Algorithm 1). Our algorithm relies on a subroutine call to any algorithm that enumerates cells in a hyperplane arrangement (Algorithm 2).

Activation regions are identified by their NSVs. We refer to a NSV up to a specific layer $k$ as *network sign vector prefix*. We refer to the set of such NSV prefixes as $V_k$.

**Definition B.1** (Network sign vector prefix). Let $\mathbf{v}$ be a network sign vector. The *network sign vector prefix* of $\mathbf{v}$ up to layer $k$, where $k \leq L$, is $\mathbf{v}[1..k] = \langle \mathbf{s}_1, \ldots, \mathbf{s}_k \rangle$.

The algorithm takes as inputs the parameters of the network, $\theta$. The first layer NSV prefixes are found by calling a cell enumeration for hyperplane arrangements subroutine (H-CellEnum; Algorithm 2) on the first layer hyperplane arrangement, $\mathcal{A}_1$. The subroutine returns all of the sign vectors

corresponding to the cells created by the arrangement. The algorithm proceeds by iterating over the rest of the hidden layers, $2 \leq l \leq L$.

---

**Algorithm 1:** NN-CellEnum

**Input:** the parameters $\theta$ of the neural network
**Output:** the set V of all network sign vectors

1 **Function** NN_CellEnum ($\theta$, $V$):
2     $V_1 := $ H_CellEnum($\mathcal{A}_1, \mathbb{R}^d$)
3     **for** $l$ in $[2, \dots, L]$ **do**
4        $V_l \leftarrow \emptyset$
5        **foreach** $\mathbf{v}[1..l-1] \in V_{l-1}$ **do**
6           $S_{\mathbf{v}[1..l-1]} := $ H_CellEnum($\mathcal{A}_l | C(\mathbf{v}[1..l-1]), C(\mathbf{v}[1..l-1])$)
7           **foreach** $\mathbf{s}_l \in S_{\mathbf{v}[1..l-1]}$ **do**
8              $V_l \leftarrow V_l \cup \{\mathbf{v} \| \mathbf{s}_l\}$
9           **end**
10        **end**
11     **end**
12     **return** $V_L$

---

**Algorithm 2:** H-CellEnum finds all cells of a hyperplane arrangement within a bounding cell.

**Input:** a hyperplane arrangement $\mathcal{A}$
       a bounding cell $B$
**Output:** the set $S$ of sign vectors of $\mathcal{A}$ within $B$

---

For each cell, $C(\mathbf{v}[1..l])$, found from the partitions formed by layers 1 to $l-1$, we construct the hyperplane arrangement, $\mathcal{A}_l | C(\mathbf{v}[1..l])$, for the neurons in layer $l$ conditioned on the effective weights and biases of cell $C(\mathbf{v}[1..l])$. We then call the cell enumeration subroutine with arrangement $\mathcal{A}_l | C(\mathbf{v}[1..l])$ and find all cells (and their respective sign vectors) of arrangement $\mathcal{A}_l | C(\mathbf{v}[1..l])$ that intersect with $C(\mathbf{v}[1..l])$. We now have found all cells formed by the network from layers 1 to $l$.

At the final layer, we have found all NSVs and thus the set of all activation regions formed by the network, $V_L$. We return $V_L$.

### B.1   Cell enumeration for a hyperplane arrangement

There are many existing algorithms for enumerating cells of a hyperplane arrangement. In this section we overview a naive algorithm (we call exhaustive enumeration) and the incremental enumeration algorithm (Rada & Černý, 2018). Another popular algorithm is the reverse search algorithm (Avis & Fukuda, 1996; Sleumer, 2000).

Let $n$ be the number of hyperplanes in the arrangement and $d$ be the dimension of the ambient space. If $n \leq d$, the arrangement is in general position, and most cells are assumed to intersect intersect with the bounded domain, then the naive exhaustive enumeration algorithm is efficient (Algorithm 3).

The exhaustive enumeration algorithm checks all possible $2^n$ sign vectors by solving a linear program to see if a witness point exists for each possible sign vector. The algorithm appears computationally expensive. However, if the number of cells formed by the arrangement is close to $2^n$ then this algorithm is output-polynomial with complexity $O(2^n \cdot \text{lp}(n, d))$, where $O(\text{lp}(n, d))$ is the time needed to formulate and solve a linear program for a witness point.

If $n > d$, there are more efficient algorithms, such as incremental enumeration (Rada & Černý, 2018) and reverse search (Avis & Fukuda, 1996; Sleumer, 2000) – which both have time complexity $O(|S|n\text{lp}(n, d))$. Here we describe the incremental enumeration algorithm (Algorithm 4). We refer to a *sign vector prefix* as $\mathbf{s}[1..k]$, i.e., the sign vector up to $k$th hyperplane.

The algorithm performs an incremental construction of the sign vectors for the cells of a given hyperplane arrangement. At iteration $k+1$, the algorithm checks if the hyperplane $H_{k+1}$ intersects

the cells described by the length $k$ sign vector prefixes that exist in the arrangement. If $H_{k+1}$ splits cell $C(\mathbf{s}[1..k])$, then two length $k+1$ prefixes are formed: $\mathbf{s}_i[1..k] \,\|\, -$ and $\mathbf{s}_i[1..k] \,\|\, +$. If the $k+1$ hyperplane falls outside the cell, then the length $k$ sign vector is extended to a length $k+1$ sign vector with the sign dependent on the orientation of $H_{k+1}$.

---

**Algorithm 3:** ExhEnum

---

**Input:** a hyperplane arrangement $\mathcal{A}$
   a bounding cell $B$
**Output:** the set $S$ of sign vectors of $\mathcal{A}$ within $B$

1 **Function** ExhEnum ($\mathcal{A}$, $B$):
2   $S \leftarrow \emptyset$
3   **foreach** $\mathbf{s} \in \{+, -\}^{|\mathcal{A}|}$ **do**
4    **if** there exists a witness point $w$ of $C(\mathbf{s}) \cap B$ **then**
5     $\mid$ $S \leftarrow S \cup \{\mathbf{s}\}$
6    **end**
7   **end**
8   **return** $S$

---

---

**Algorithm 4:** IncEnum

---

**Input:** a hyperplane arrangement $\mathcal{A}$
   a bounding cell $B$
**Output:** the set $S$ of sign vectors of $\mathcal{A}$ within $B$

1 **Function** IncEnum ($\mathcal{A}$, $B$):
2   $S \leftarrow \emptyset$
3   $w \leftarrow$ a witness point of $B$
4   _IncEnum ($\langle\rangle$, $w$)
5   **return** $S$
6 **Function** _IncEnum ($\mathbf{s}[1..k]$, $w$):
7   **if** $\mathbf{s}[1..k]$ is empty **then** $k \leftarrow 0$
8   **if** $k < n$ **then**
9    $\sigma \leftarrow$ **if** $w \in H_{k+1}^+$ **then** $+$ **else** $-$
10    _IncEnum ($\mathbf{s}[1..k]\|\sigma$, $w$)
11    **if** there exists a witness point $w'$ of $C(\mathbf{s}[1..k]\| - \sigma) \cap B$ **then**
12     $\mid$ _IncEnum ($\mathbf{s}[1..k]\| - \sigma$, $w'$)
13    **end**
14   **end**
15   **else**
16    $\mid$ $S \leftarrow S \cup \{\mathbf{s}[1..n]\}$
17   **end**

---

If one chooses conditions wisely on which cell enumeration subroutine to call, then the `NN-CellEnum` algorithm will itself be output-polynomial since there are $O(L|V|)$ number of calls to the `H-CellEnum` subroutines.

## C NEIGHBOR FINDING ALGORITHM PSEUDO CODE

Here we present the pseudo code for our neighbor finding algorithm, `NN_Nieghbors`. The core idea of the approach is adapted from Sleumer (2000) for finding tight hyperplanes in a hyperplane arrangement. Algorithm 5 is the pseudo code for the neighbor finding algorithm in its entirety, and Algorithm 6 is the pseudo code for the subroutine that finds the remaining tight hyperplanes.

---

**Algorithm 5:** NN_Neighbors.

---

**Input:** the parameters $\theta$ of the neural network
the set $V$ of network sign vectors
**Output:** the set $E$ of network sign vector pairs of neighboring regions

1 **Function** NN_Neighbors($\theta$, $V$):
2   $E \leftarrow \{\}$
3   **foreach** $\mathbf{v} \in V$ **do**
4    $\mathcal{A}_\mathbf{v} \leftarrow$ NetHypArr($\theta$, $\mathbf{v}$)      // network hyp arr for cell $C(\mathbf{v})$
5    $\mathbf{p} \leftarrow$ IntPt($\mathcal{A}_\mathbf{v}$, $\mathbf{v}$)       // interior point of $C(\mathbf{v})$ in $\mathcal{A}_\mathbf{v}$
6    $\mathcal{T}_0 \leftarrow$ InitTightHyp($\mathcal{A}_\mathbf{v}$, $\mathbf{v}$)     // initial tight hyps of $C(\mathbf{v})$
7    $\mathcal{T} \leftarrow$ TightHyp($\mathcal{A}_\mathbf{v}$, $\mathbf{v}$, $\mathbf{p}$, $\mathcal{T}_0$)    // all tight hyps of $C(\mathbf{v})$
8    **foreach** $T_{l,i} \in \mathcal{T}$ **do**
9     $\mathbf{v}' \leftarrow \mathbf{v}$ with $v'_{l,i} := -v_{l,i}$
10     $E \leftarrow E \cup \{\{\mathbf{v}, \mathbf{v}'\}\}$
11    **end**
12   **end**
13   **return** $E$

---

**Algorithm 6:** TightHyperplanes

---

**Input:** the network hyperplane arrangement $\mathcal{A}_\mathbf{v}$ for cell $C(\mathbf{v})$
the network sign vector $\mathbf{v}$ corresponding to cell $C(\mathbf{v})$
an interior point $\mathbf{p}$ of $C(\mathbf{v})$
a vertex defining set of tight hyperplanes $\mathcal{T}_0$ in $\mathcal{A}_\mathbf{v}$
**Output:** the set $\mathcal{T}$ of tight hyperplanes of $C(\mathbf{v})$

1 **Function** TightHyp($\mathcal{A}_\mathbf{v}$, $\mathbf{v}$, $\mathbf{p}$, $\mathcal{T}_0$):
2   $\mathcal{U} \leftarrow \mathcal{A}_\mathbf{v} \setminus \mathcal{T}_0$
3   **while** $|\mathcal{U}| \neq 0$ **do**
4    select a $H_{l,i} \in \mathcal{U}$
5    **if** $W_\mathbf{q}(H_{l,i})$ **then**
6     find $H_{k,j} \in \mathcal{U}$ that intersects ray $\overline{\mathbf{pq}}$ closest to $\mathbf{p}$
7     $\mathcal{T} \leftarrow \mathcal{T} \cup \{H_{k,j}\}$; $\mathcal{U} \leftarrow \mathcal{U} \setminus \{H_{k,j}\}$
8    **else**
9     $\mathcal{U} \leftarrow \mathcal{U} \setminus \{H_{l,i}\}$
10    **end**
11   **end**
12   **return** $\mathcal{T}$

---

# D   TIME COMPLEXITY ANALYSIS

## D.1   CELL ENUMERATION TIME COMPLEXITY

The time complexity of the cell enumeration algorithm is $O(L|V|n\mathrm{lp}(n,d))$, which is linear in the size of the output, $V$. Assume the use of an efficient subroutine, such as IncEnum or reverse search (Avis & Fukuda, 1996; Rada & Černý, 2018). The time to find the new cells formed by layer $l$ in a cell, $C(\mathbf{v}[1..l-1])$, formed by the previous layers is $O(|S_{\mathbf{v}[1..l-1]}|n_l\mathrm{lp}(n_l,d))$, where $S_{\mathbf{v}[1..l-1]}$ is the set of sign vectors found within $C(\mathbf{v}[1..l-1])$. The time to find all new regions formed by layer $l$ is $O(|V_l|n_l\mathrm{lp}(n_l,d))$. Iterating over all layers gives the final time complexity.

## D.2   NEIGHBOR FINDING TIME COMPLEXITY

The total time complexity of neighbor finding is $O(|V|n\mathrm{lp}(n,d) + |V|n^2)$. For a given $\mathbf{v} \in V$, the runtime is dominated by the step for finding the remaining tight hyperplanes (Algorithm 6 and line 7 of Algorithm 5). In each iteration a linear program of complexity $O(\mathrm{lp}(n,d))$ is ran to find point $\mathbf{q}$ for constructing the ray $\overline{\mathbf{pq}}$. If $\mathbf{q}$ exists, each hyperplane in $\mathcal{U}$ is tested to be the first hit by $\overline{\mathbf{pq}}$, taking $O(n)$ time. Each iteration removes one hyperplane from the $\mathcal{U}$. So the total time of the

function call to find the remaining tight hyperplanes is $O(n(\text{lp}(n, d) + n))$, and the function is called for all regions in $V$.

# E  CORRECTNESS ARGUMENT

## E.1  CELL ENUMERATION CORRECTNESS ARGUMENT

We show that Algorithm 1 finds all activation regions formed by the network. We provide an inductive proof for the following property.

**Property 1.** For layer $l$ of the network, our algorithm finds the partition regions formed by layers 1 to $l$ of the network and corresponding NSV prefixes, $V_l$.

First we show that the subroutines (H-CellEnum) we use, ExhEnum (Algorithm 3) and IncEnum (Algorithm 4), find all cells in a given hyperplane arrangement. For the correctness of IncEnum see Rada & Černý (2018). In ExhEnum, all possible $2^{|\mathcal{A}|}$ sign vectors are tested. If an interior point exists, the cell exists and the sign vector is included in the returned set.

We now prove Property 1 by induction.

Base case 1 ($l = 1$): H-CellEnum is called on $\mathcal{A}_1$ and returns all sign vectors $V_1$.

Inductive step: Assume Property 1 holds when $1 \leq k < L - 1$, we show it holds for $k + 1$. For each NSV prefix in $V_k$, run a H-CellEnum subroutine. Without loss of generality, let $C(\mathbf{v}[1..k])$ be the partition region of $\mathbf{v}[1..k] \in V_k$. H-CellEnum returns the set of sign vectors, $S_{\mathbf{v}[1..k+1]}$, that correspond to the cells in the hyperplane arrangement $\mathcal{A}_{k+1}|C(\mathbf{v}[1..k])$ that intersect with $C(\mathbf{v}[1..k])$. Appending the sign vectors in $S_{\mathbf{v}[1..k+1]}$ to $\mathbf{v}[1..k]$ forms a set of NSV prefixes. The union of these sets is the set of all NSV prefixes from layers 1 to $k + 1$, $V_{k+1}$.

At the final layer, the algorithm has found $V_L$ which is the set of all activation regions, $V$.

## E.2  NEIGHBOR FINDING CORRECTNESS ARGUMENT

The tight hyperplanes of $C(\mathbf{v})$ in $\mathcal{A}_\mathbf{v}$ are the bounding hyperplanes of $C(\mathbf{v})$ in the tessellation formed by the DNN. We show that lines 4-11 of Algorithm 5 finds exactly all tight hyperplanes $C(\mathbf{v})$ in $\mathcal{A}_\mathbf{v}$ and thus all neighbors.

Assume $\mathcal{A}_\mathbf{v}$, $\mathbf{p}$, and $\mathcal{T}_0$ have been correctly computed. We show that the addition of a hyperplane to $\mathcal{T}$ is always a tight hyperplane. Let $H_{l,i}$ be the hyperplane being added to $\mathcal{T}$. Assume for the sake of contradiction that $H_{l,i}$ is not a tight hyperplane of $C(\mathbf{v})$ in $\mathcal{A}_\mathbf{v}$. Then the ray $\overline{\mathbf{pq}}$ intersects at least one other hyperplane before $H_{l,i}$. This is a contradiction of the definition of a tight hyperplane. Thus, $H_{l,i}$ must be tight to $C(\mathbf{v})$.

We now show that the algorithm finds all tight hyperplanes of $C(\mathbf{v})$. For the sake of contradiction, assume $H_{k,j}$ is a tight hyperplane not included in the set of tight hyperplanes returned by the algorithm. Then the predicate $W_\mathbf{q}(H_{k,j})$ was false, i.e., a witness $\mathbf{q}$ does not exist for arrangement $\mathcal{T}_\delta$. So, there is no point that satisfies the constraints of $\mathcal{T}$ and lies on the hyperplane $H_{k,j}$. Thus, $H_{k,j}$ is not a tight hyperplane of $\mathcal{T}$ and therefor not a tight hyperplane of $C(\mathbf{v})$ – a contradiction.

# F  IMPLEMENTATION DETAILS AND VALIDATION

All code was written in Python 3.10. Gurobi (Optimization, 2014) was used as the linear program solver. We set the feasibility tolerance to the minimum value allowed by Gurobi ($10^{-9}$) and found no constraint violations in all of our experiments.

To further test for numerical issues, we ran our algorithm on models trained on the two-spiral problem (Figure App. 6). We found that extremely small and extremely large effective weights and biases resulted in errors in our algorithm's output, despite not violating any constraints in the linear program solver. Scaling these parameters to a more appropriate range (without changing the network hyperplane arrangements) resolved the issues. Despite these issues, in our models trained on EMNIST-15, no such scaling was necessary since the effective weights and biases were well within the range for our algorithm to perform accurately.

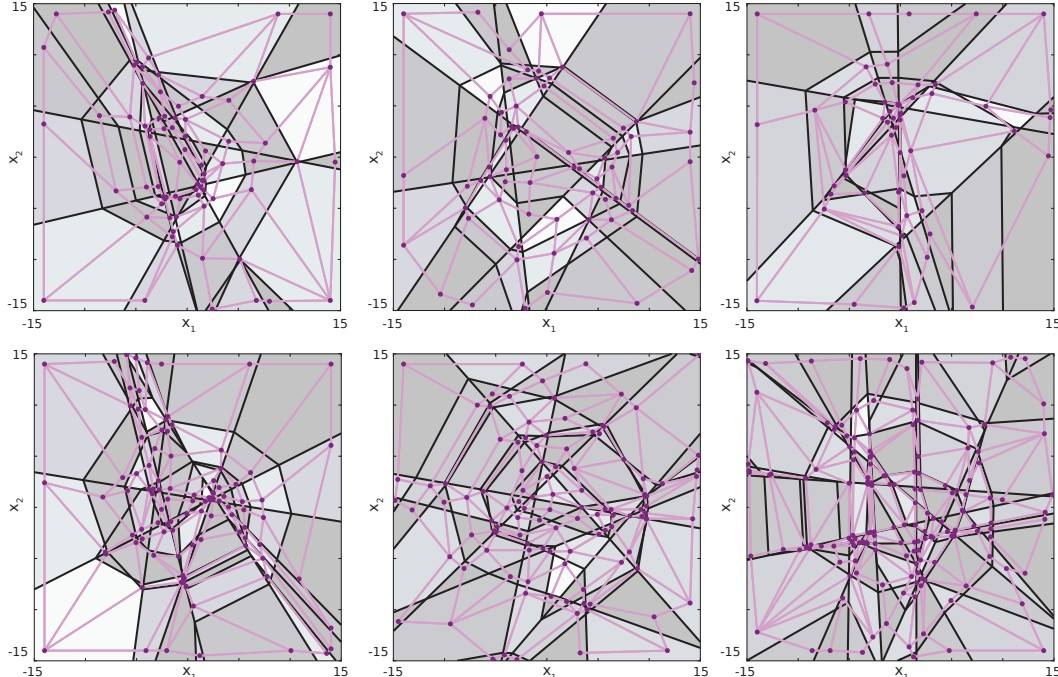

Figure App. 6: Network partition regions (gray) and corresponding dual graphs (purple) found by our algorithm on models trained on the two-spiral problem. Top: Networks with architectures $n_1 = n_2 = 5$. Bottom: Networks with architectures $n_1 = n_2 = 6$, $n_1 = n_2 = 7$, and $n_1 = n_2 = 8$, from left to right.

In our implementation, we parallelized both the cell enumeration and neighbor finding processes using a master-worker paradigm with RabbitMQ as the message broker. For cell enumeration, since $n_1 << d$ in our experiments, a unit of work comprised of determining if a specific first layer NSV prefix $\mathbf{v}[1]$ existed; if it existed then the worker used IncEnum to find all partitions formed by the deep layers formed within the first layer partition corresponding to $\mathbf{v}[1]$. We did this for all possible 2,048 first hidden layer NSV prefixes. In other words, we performed ExhEnum to find partitions in the first hidden layer and IncEnum for the deep layer.

For neighbor finding, each cell found by cell enumeration contributed a unit of work. Workers ran our neighbor finding algorithm (lines 4 through 11 in Algorithm 5) to find all neighbors for a given input cell. All reported runtimes are the aggregate time across all workers, as if the algorithm ran sequentially.

