# OpenReview forum: "Building the dual graph of the activation regions in a deep neural network: what it means for interpretability"
_ICLR.cc/2026/Conference — Submitted to ICLR 2026_

### Official Review · Reviewer_bhka · 2025-10-25

**Soundness:** 2
**Presentation:** 3
**Contribution:** 2
**Rating:** 2
**Confidence:** 4

**Summary:**

This paper explores the geometric representations of deep neural networks by introducing a method to compute the dual graph of the input space partition induced by a continuous piecewise affine deep network on its input space. The authors then implement this method to compute the dual graphs of deep neural networks trained on the Extended MNIST dataset. They find that measuring the paths being training samples along the dual graph provides insights into the generalisation of the deep neural network.

**Strengths:**

1. Well Motivated: The challenge of computing the dual graph of an input space partition is well motivated, and the proposed method is outlined clearly.
2. Novel Approach that Builds Upon Existing Work: The authors effectively leverage prior results/algorithms where possible, but then derive a novel neighbour finding algorithm that extends existing work.

**Weaknesses:**

1. Correctness of Algorithm: The proposed method is described; however, no formal statement or validation is provided on a toy example.
2. Small Experiments and Practical Scalability: As deep networks become large, the number of activation regions grows exponentially; therefore, computing the dual graph would seem intractable. Furthermore, supposing obtaining a dual graph is feasible, algorithms operating on these graphs are likely to be computationally expensive. Perhaps evidenced by the fact that only small-scale experiments are considered.
3. Overstating the Importance and Applicability to Interpretability: Using the term 'interpretability' is somewhat misleading. Interpretability is typically used in reference to understanding the learned features of a deep network. Although this application is mentioned in the conclusion, it is not the main focus or application considered in the paper.

**Questions:**

1. What is the computational complexity of the proposed method for finding the dual graph?
2. How would you consider adapting the algorithm for deep networks that are not continuous piecewise affine?
3. How does the utilisation of this exact algorithm compare to just sampling points along a linear interpolation of input points and using the number of unique input-output Jacobians as a proxy for the shortest path?
4. Is there a property of a deep neural network that you foresee we can only explain using the dual graph, rather than just using the input space partition?

---

> ### Author Response · Authors · 2025-11-28
>
> We thank the reviewer for the constructive feedback and thought provoking questions, specifically the questions regarding the importance of the dual graph structure and benefit over existing sampling methods. We address their comments here:
>
> [W1] Correctness of Algorithm: The proposed method is described; however, no formal statement or validation is provided on a toy example.
>
> We have included both validation on a toy problem (Appendix F and Figure App. 6) and a correctness argument (Appendix E). The two-spiral problem is used as a toy problem to validate the algorithm.
>
> [W2] Small Experiments and Practical Scalability: As deep networks become large, the number of activation regions grows exponentially; therefore, computing the dual graph would seem intractable. Furthermore, supposing obtaining a dual graph is feasible, algorithms operating on these graphs are likely to be computationally expensive. Perhaps evidenced by the fact that only small-scale experiments are considered.
>
> The reviewer is correct that scaling to larger networks is computationally expensive and in fact intractable for the models used in applications today. However, we do believe that there is utility in constructing the exact dual graph for small problems to study the geometry of deep networks.
>
> To extend beyond small models, we also present approximate methods for neighbor finding and shortest paths (Section 6.2). The accuracy of these methods were evaluated in the EMNIST models and found to have very few errors (Table 2). These methods are relatively quick and can be applied to larger problems. Specifically, we apply the approximate methods to a CNN trained on CIFAR-10, and we find the approximate dual graph shortest paths in the CNN are more predictive of generalization than Euclidean distance (Figure 5).
>
> [W3] Overstating the Importance and Applicability to Interpretability: Using the term 'interpretability' is somewhat misleading. Interpretability is typically used in reference to understanding the learned features of a deep network. Although this application is mentioned in the conclusion, it is not the main focus or application considered in the paper.
>
> We have edited the manuscript for usage of such language. We now use the term “model explainability” in place of “interpretability.”

---

> ### Author Response · Authors · 2025-11-28
>
> [Q1] What is the computational complexity of the proposed method for finding the dual graph?
>
> We now include a time complexity analysis (Appendix D) as well as empirical runtimes (Table 1).
>
> [Q2] How would you consider adapting the algorithm for deep networks that are not continuous piecewise affine?
>
> This is a great question. While we do not have a straightforward answer, we do wish to adapt our perspective to non continuous piecewise affine networks as well.
>
> [Q3] How does the utilisation of this exact algorithm compare to just sampling points along a linear interpolation of input points and using the number of unique input-output Jacobians as a proxy for the shortest path?
>
> This is an interesting question and an experiment worth running as future work. The paper now has three relevant adjustments pertaining to this question.
>
> First, we include and analyze the accuracy of Hamming distance between network sign vectors as an approximate method for measuring the shortest path in the dual graph (as described in [W2]; Section 6.2).
>
> Second, we include in the discussion section (Section 7) the future work direction of comparing Hamming distance to the input-output Jacobian norm and cite the relevant work by Novak et al. (2018).
>
> Finally, we include another future work direction inspired by the input-output Jacobian norm perspective suggested by the reviewer. Specifically, we are interested in analyzing the change in affine transformations of partition regions as a function of path distance in the neighborhood around a training point. This analysis would require constructing a small subgraph around an input point, which could be done with approximate neighbor finding (Section 6.2).
>
> [Q4] Is there a property of a deep neural network that you foresee we can only explain using the dual graph, rather than just using the input space partition?
>
> We have many future direction ideas that would require both the edges and vertices of the dual graph, which is now reflected in the discussion section. Specifically, any analysis that requires a measure of distance or discovering a neighborhood around an input, such as: (1) the relationship between features (as cuts in the graph) and their stability to class boundaries (as walks); (2) the correlation between bin distance and generalization and bin distance and image morphology (Balestriero & Baraniuk 2018); and (3) the piecewise linear network representations in a neighborhood around an input point as a function of path length in the graph.

---

### Official Review · Reviewer_oZdQ · 2025-10-29

**Soundness:** 2
**Presentation:** 3
**Contribution:** 3
**Rating:** 4
**Confidence:** 4

**Summary:**

This paper introduces the first exact algorithm for constructing the dual graph of activation regions in deep neural networks (DNNs) with piecewise linear activations (e.g., ReLU). Unlike prior work that focuses solely on enumerating activation regions, this work identifies neighboring regions—a much harder problem due to conditional layer dependencies. The authors leverage results from computational geometry (e.g., Sleumer 2000, Rada & Černý 2018) to design an output-polynomial algorithm that enumerates regions and discovers neighbors via "tight hyperplane" detection. They then demonstrate the interpretive utility of the dual graph by correlating graph-based path distances between training and test regions with generalization performance on EMNIST.

**Strengths:**

* The proposed work has its novelty in that the authors are the first to explicitly formalize and solve the neighbor-finding problem for DNN activation regions. Prior work cited in Related Work (p. 2 L107–L141) enumerates cells but not adjacency—this paper extends the geometric picture to the full dual graph.

* The construction of the network hyperplane arrangement and the use of linear programs for interior-point search are mathematicallly consistent with computational geometry practice.

**Weaknesses:**

* The paper claims the algorithm is output-polynomial (pg. 6) but gives no empirical runtime or complexity scaling.
* Experiments seem to be confined to 2-layer MLPs with width = 11. There is no timing or memory analysis across larger architectures.
* Adding the comparative evaluation based on some of the related works (e.g. Balestriero & LeCun (2024)) would strengthen the work. Without such comparisons, it is unclear if the new algorithm improves over existing enumeration methods
* The observed correlation between path length and accuracy (Figure 6) is very interesting but not anlayzed in depth. The discussion section uses strong claims ("operationally meaningful definition of generalization") without rigorous justification.

The dual-graph formalism represents a meaningful conceptual and algorithmic advance over prior geometric analyses of neural networks.
However, the work remains computationally limited and empirically under-demonstrated (Sections 5–6, p. 6–8).
If scalability evidence and stronger experimental validation are added, I would be happy to raise the score.

** Some minor points **
* Sometimes the input is denoted as $\theta$ but sometimes it's $\mathbf{x}$
* Minor typographical inconsistencies (e.g., “tesselation” (pg. 8) vs. “tessellation” (pg. 6)) and overuse of the term “network sign vector” without shorthand notation make some passages verbose.
* The “bounded domain” discussion (Appendix D) could be integrated earlier for completeness.
* References to recent theoretical works on piecewise-linear region geometry (e.g., Raghu et al. 2017, Poole et al. 2016) are missing.

**Questions:**

* Please provide empirical scaling results (runtime vs. number of neurons/layers) and compare them with prior region enumeration methods.
* Approximation or Sampling: Can the proposed algorithm be adapted into an approximate dual-graph construction for larger models? If so, outline how accuracy vs. computational cost would trade off.
* Beyond the EMNIST correlation, can the dual graph capture interpretable clusters (e.g., digits with similar morphology) or identify adversarial transitions across edges?
* Does the “output-polynomial” complexity claim hold under degenerate network configurations where many neurons are inactive or redundant?
* How sensitive is the neighbor-finding step to numerical precision (e.g., floating-point stability in LP solvers)?

---

> ### Author Response · Authors · 2025-11-28
>
> We thank the reviewer for the very thoughtful and constructive comments, especially the comments pertaining to empirical runtime, algorithm validation, linear program precision, and approximate methods. We address their comments here:
>
> [W1] The paper claims the algorithm is output-polynomial (pg. 6) but gives no empirical runtime or complexity scaling.
>
> We include both empirical runtimes (Table 1) and a complexity analysis (Appendix D).
>
> [W2] Experiments seem to be confined to 2-layer MLPs with width = 11. There is no timing or memory analysis across larger architectures.
>
> To extend beyond 2-layer MLPs with width=11, we now include approximate methods for both neighbor finding and the path length generalization analysis (Section 6.2 and Figure 5). We run these methods on a CNN with 5 convolutional layers and two fully connected layers trained on CIFAR-10.
>
> We are also now running the exact algorithm on other 2-layer and deeper MLPs, which we would be happy to include in the presentation.
>
> [W3] Adding the comparative evaluation based on some of the related works (e.g. Balestriero & LeCun (2024)) would strengthen the work. Without such comparisons, it is unclear if the new algorithm improves over existing enumeration methods
>
> We have included a comparison between our work and Balestriero & LeCun (2024), see Section 5.1. Specifically, we state that Balestriero & LeCun (2024) implement a variation of IncEnum which does not return network sign vectors, and that our work extends theirs by implementing IncEnum, albeit directly so that we can use the network sign vector outputs as inputs into the neighbor finding algorithm. We also emphasize that the novelty of our contribution is the neighbor finding step, which extends region enumeration.
>
> Unfortunately their paper does not provide details for the parallel implementation or a theoretical bound beyond saying “our method has linear asymptotic complexity with respect to the number of regions and with respect to the DN’s input space dimension.” Thus, it is hard to make a direct comparison, but our algorithm is also linear with respect to the number of regions, the input space dimension, and the number of neurons. Further, their empirical results are restricted to only shallow networks with fewer regions. Our work extends to networks with deep layers and many more regions as well as approximate methods for larger networks.
>
> We now include time complexity of our algorithms (Appendix D) so that it can be better compared to other work.
>
> [W4] The observed correlation between path length and accuracy (Figure 6) is very interesting but not analyzed in depth. The discussion section uses strong claims ("operationally meaningful definition of generalization") without rigorous justification.
>
> We have revised the discussion section to avoid the strong claims that were previously written.
> Currently, we are further analyzing path lengths and test accuracies, as well as the graph structure and its relationship to other properties of network performance. We invite any feedback on interesting future research questions/directions.
>
> Minor points: We have addressed all the minor points. We now include the “bounded domain” discussion in the main text (see the end of Section 4.1). We also now include the references that the reviewer has suggested.

---

> ### Author Response · Authors · 2025-11-28
>
> [Q1]: Please provide empirical scaling results (runtime vs. number of neurons/layers) and compare them with prior region enumeration methods.
>
> We now include empirical scaling results in Table 1.
>
> As discussed in response to [W3], we cannot provide a rigorous comparison to Balestriero & LeCun (2024).
>
> [Q2]: Approximation or Sampling: Can the proposed algorithm be adapted into an approximate dual-graph construction for larger models? If so, outline how accuracy vs. computational cost would trade off.
>
> We now include an approximate method for neighbor finding, and we show that the Hamming distance between two network sign vectors is a good approximation for shortest path length in the dual graph. We detail the accuracy of these methods in Table 2 and describe how they work in Section 6.2.
>
> [Q3]: Beyond the EMNIST correlation, can the dual graph capture interpretable clusters (e.g., digits with similar morphology) or identify adversarial transitions across edges?
>
> We think these are very interesting ideas that we hope to explore in the future.
>
> Balestriero & Baraniuk (2018) did a morphology experiment using the approximate distance method that we use in Figure 4. They found that in deeper layers of networks, images with small Hamming distances of network sign vectors become closer in feature similarity but further in Euclidean distance (Figure 4). We now include their result in the discussion section of our paper (Section 7).
>
> [Q4]: Does the “output-polynomial” complexity claim hold under degenerate network configurations where many neurons are inactive or redundant?
>
> The only assumption we require is that the network hyperplane arrangements are in general position.
> This assumption does not hold true if the effective weights and biases for two neurons are the same. This degeneracy would occur if all the weights from the inputs to the two neurons being compared are exactly the same. In other words, if two neurons form the exact same cut in a region of the input space (i.e., the same hyperplane in a network hyperplane arrangement), our algorithm would fail. However, we believe the algorithm could be adapted for such edge cases, either by identifying redundancies or using perturbation methods (as done in Sleumer 2000), but in practice such an edge case is unlikely to occur.
>
> The number of active or inactive neurons does not impact our algorithm (unless a neuron is inactive because all weights are 0). In fact, the algorithm may run faster for networks with many inactive neurons as this would decrease the dimensionality of the function within the partition regions, likely improving the runtime of the linear program calls and reducing the number of partitions formed by deeper layers.
>
> [Q5]: How sensitive is the neighbor-finding step to numerical precision (e.g., floating-point stability in LP solvers)?
>
> We include toy validation experiments to address this question (Appendix F and Figure App. 6). We find that very large or very small effective weights and biases cause numerical precision errors in models trained on the two-spiral problem. Scaling the coefficients of the violating hyperplanes in the hyperplane arrangement resolves these issues. However, for the EMNIST models, the effective weights and biases are well within the range where the algorithm performed accurately on the two-spiral problem.

---

> ### Author Response · Authors · 2025-11-28
>
> We hope that we have addressed the concerns of scalability evidence by including a complexity analysis (Appendix D), empirical runtimes (Table 1), and approximate methods applied to a larger network (CNN) with a richer dataset (CIFAR-10; Table 2). And we hope that we have addressed the concern of experimental validation by analyzing path length and generalization on the larger network (Figure 5).

---

### Official Review · Reviewer_CgHs · 2025-10-30

**Soundness:** 3
**Presentation:** 2
**Contribution:** 3
**Rating:** 6
**Confidence:** 2

**Summary:**

The paper studies the full geometric representation of piecewise-linear deep neural networks by building dual graphs. The authors show that finding neighbors in deep networks is nontrivial because of the conditional, layer-wise partitioning that makes adjacency harder to detect than in single-layer hyperplane arrangements. They present what they claim is the first exact algorithm to construct the dual graph for such DNNs, and combine it with a region-enumeration procedure to recover the complete graph. Using this graph, they define a path-length distance on the manifold of activation regions and empirically show that test points that are close to training points under this metric tend to generalize better.

**Strengths:**

1. The paper identifies the dual graph of activation regions, moving beyond mere enumeration of regions to their adjacency structure.

2. It provides an exact algorithm for neighbor finding in deep networks, which is harder than commonly assumed.

3. The dual-graph viewpoint is intuitive and positions path length as a natural, geometry-aware metric that can capture relationships missed by Euclidean distances in input space.

**Weaknesses:**

1. A few sentences are slightly awkward and could be tightened for clarity: “This may be due to a naive assumption that because identifying neighboring regions is trivial in shallow networks, it too is trivial in deep networks.” reads as clunky.

2. Experiments are restricted to relatively simple, shallow architectures and MNIST. It is unclear how the enumeration + neighbor-finding pipeline scales to practical modern networks (e.g., deeper convnets, transformers, large input dimensions) or to richer datasets; the paper should either provide evidence of scalability or clearly delimit the claimed scope.

**Questions:**

1. I am not a specialist in every low-level technical aspect of region enumeration algorithms, so I cannot fully vouch for implementation subtleties or edge cases. In particular, I would like the authors to explain the broader significance of studying the dual graph for DNN theory and practice, given their focus on simple networks. Why should practitioners or theorists prioritize this object, and what concrete insights does it unlock beyond what regions already provide?

---

> ### Author Response · Authors · 2025-11-28
>
> We thank the reviewer for the thoughtful feedback, especially the comment pertaining to the importance of the dual graph structure. We address their comments here:
>
> [W1] A few sentences are slightly awkward and could be tightened for clarity: “This may be due to a naive assumption that because identifying neighboring regions is trivial in shallow networks, it too is trivial in deep networks.” reads as clunky.
>
> We have revised the manuscript to avoid clunkiness.
>
> [W2] Experiments are restricted to relatively simple, shallow architectures and MNIST. It is unclear how the enumeration + neighbor-finding pipeline scales to practical modern networks (e.g., deeper convnets, transformers, large input dimensions) or to richer datasets; the paper should either provide evidence of scalability or clearly delimit the claimed scope.
>
> We now clearly delimit the scope of the exact algorithm. We include both empirical runtimes (Table 1) and a complexity analysis (Appendix D).
>
> Further, we now include approximate methods to scale the dual graph construction and generalization analysis to a larger network with a richer dataset, specifically a CNN trained on CIFAR-10 (Section 6.2 and Figure 5). We analyze the accuracy of our approximate neighbor finding and shortest path methods on our original three models trained on EMNIST (Table 2).
>
> [Q1] I am not a specialist in every low-level technical aspect of region enumeration algorithms, so I cannot fully vouch for implementation subtleties or edge cases. In particular, I would like the authors to explain the broader significance of studying the dual graph for DNN theory and practice, given their focus on simple networks. Why should practitioners or theorists prioritize this object, and what concrete insights does it unlock beyond what regions already provide?
>
> Regions alone is an incomplete description of the piecewise linear structure. This is the first complete geometric description for a deep neural network.
>
> The discussion has been modified to include future analysis directions that would not be possible with only regions. Specifically, we plan to analyze (1) the relationship between features and their stability to outcomes across the input space via the dual graph representation – features can be represented as cuts in the graph and class boundaries as walks; (2) the correlation between bin distance and generalization and bin distance and image morphology (Balestriero & Baraniuk 2018); and (3) the piecewise linear network representations in a neighborhood around an input point as a function of path length in the graph.

---

### Official Review · Reviewer_qBKY · 2025-11-01

**Soundness:** 3
**Presentation:** 2
**Contribution:** 3
**Rating:** 6
**Confidence:** 2

**Summary:**

This paper focuses on the interpretability approach of activation polytopes induced by the ReLU activations within a network, and describes a method for not only finding such regions, but for constructing a graph to describe the full set of regions and their connectivity to one another.

**Strengths:**

- Lots of technical detail

**Weaknesses:**

- The paper didn't do a good job of clearly articulating why and how activation polytopes could be used for interpretability
- The paper focused too much on minutia that is confusing to a non-expert, and not enough on the ways that such a technique could be used

**Questions:**

- How well does this method scale to larger networks?

---

> ### Author Response · Authors · 2025-11-28
>
> We thank the reviewer for the helpful feedback. We address the reviewer’s comments here:
>
> [W1]: The paper didn’t do a good job of clearly articulating why and how activation polytopes could be used for interpretability.
>
> Prior work has demonstrated that understanding the polytope activation regions formed by ReLU networks is a promising direction for understanding network performance.
>
> Specifically, some previous results are the following. The number of regions correlates with network performance/expressivity (Hanin & Rolnick 2019). The density of regions around data points and the variance of the affine functions within the polytopes are indicative of network generalizability (Humayun et al. 2024; Novak et al. 2018). Further, visualization tools have been useful for gleaming insight into how architecture choices and training procedures change polytope organization and structure (Balestriero et al. 2022; Humayun et al. 2023).
>
> [W2]: The paper focused too much on minutia that is confusing to a non-expert, and not enough on the ways such a technique could be used.
>
> While in this paper we focused on the algorithm and its novelty, we do also show an application of the technique for model generalization.
> One of our results shows how the shortest paths in the graph are predictive of generalization of unseen test data (Figures 4 and 5). In the discussion, we have proposed future directions for applying and extending the dual graph as an analytical tool.
>
> [Q1]: How well does this method scale to larger networks?
>
> We include both empirical runtimes (Table 1) and a complexity analysis (Appendix D).
> To scale the technique to larger models, we include approximate methods for both finding neighboring regions and computing the shortest path length between regions. We apply the approximate path distance method to a larger model – a CNN trained on CIFAR-10. See Section 6.2, Table 2, and Figure 5.

---

### Meta-Review · Area_Chair_pknD · 2026-01-09

**Summary:**

This paper presents an exact algorithm for constructing the dual graph of activation regions in deep piecewise-linear networks, addressing the nontrivial problem of neighbor finding and demonstrating the utility of graph-based distances for analyzing generalization. Reviewers agree that the technical contribution—formalizing and solving the dual-graph construction problem—is novel and well motivated, and several concerns raised in the initial reviews were substantively addressed in the rebuttal, including added runtime analysis, correctness validation, approximate scaling methods, clearer scoping of claims, and improved presentation. However, the work remains borderline for the following reasons: the exact algorithm is still limited to small models, the approximate methods and generalization results, while promising, are not yet sufficiently mature to establish strong practical impact, and the interpretability (now rephrased) implications remain somewhat underdeveloped. At the moment, in my current assessment, I feel the extent of changes introduced during rebuttal and the remaining open questions around scalability and empirical significance would warrant another iteration and review cycle. I therefore recommend rejection at this point, with encouragement to resubmit after further consolidation and validation.

**Reviewer Concerns:**

Concerns largely addressed by the rebuttal include the added complexity analysis, empirical runtimes, and approximate methods; demonstrated on a larger CNN (CIFAR-10). Furthermore, the authors softened claims about generalization and interpretability; improved wording and scope delimitation. Concerns still outstanding include convincing practical relevance to modern deep models. Also, while terminology was fixed, concrete interpretability use-cases remain underdeveloped. Finally, I think there is need for for further experimental consolidation. Overall, the number and scope of post-rebuttal changes suggest the paper is still evolving and would benefit from another full review cycle.

**Reviewer Scores:**

While the provided rebuttal appears to resolve certain concerns, I do not think that review scores would change substantially. Especially, the "Reject" decision is very unlikely to be significantly bumped up. With only small score increases, the paper would still be borderline and, most importantly (in my opinion), the amount of changes required are substantial enough to go through another round of reviews.

---

### Decision · Program_Chairs · 2026-01-26

Reject